# A randomised Trial of Autologous Blood products, leukocyte and platelet-rich fibrin (L-PRF), to promote ulcer healing in LEprosy: The TABLE trial

**Indra B. Napit** [1,2]*, **Dilip Shrestha**[1], **Sopna Choudhury**[3], **Eleni Gkini**[4], **Onaedo Ilozumba**[3], **Paramjit Gill**[2], **Jon Bishop**[4], **Karuna Neupane**[1], **Anju Adhikari**[1], **Jo Sartori**[3], **Samuel I. Watson**[3◉], **Richard Lilford**[3◉]

**1** Anandaban Hospital, The Leprosy Mission Nepal, Kathmandu, Nepal, **2** Division of Health Sciences, Warwick Medical School, University of Warwick, Coventry, United Kingdom, **3** Institute of Applied Health Research, College of Medical and Dental Sciences, University of Birmingham, Edgbaston, Birmingham, United Kingdom, **4** Birmingham Clinical Trials Unit, College of Medical and Dental Sciences, University of Birmingham, Edgbaston, Birmingham, United Kingdom

◉ These authors contributed equally to this work.
* indran@tlmnepal.org

**Data Availability Statement:** The data underlying the findings would require participant level trial data being made publicly available which could

## Abstract

### Introduction

Autologous blood products like Platelet Rich Plasma (PRP) and Leukocyte and Platelets Rich Fibrin (L-PRF) have been used for many years across many types of skin ulcers. However, the effectiveness of autologous blood products on wound healing is not well established.

### Methods

We evaluated the 'second generation' autologous product- Leukocyte and Platelet- Rich Fibrin (L-PRF). Our trial was undertaken on patients suffering from neuropathic leprosy ulcers at the Anandaban hospital which serves the entire country of Nepal. We conducted a 1:1 (n = 130) individually randomised trial of L-PRF (intervention) vs. normal saline dressing (control) to compare rate of healing and time to complete healing. Rate of healing was estimated using blind assessments of ulcer areas based on three different measurement methods. Time to complete healing was measured by the local unblinded clinicians and by blind assessment of ulcer images.

### Results

The point estimates for both outcomes were favourable to L-PRF but the effect sizes were small. Unadjusted mean differences (intervention vs control) in mean daily healing rates (cm$^2$) were respectively 0.012 (95% confidence interval 0.001 to 0.023, p = 0.027); 0.016 (0.004 to 0.027, p = 0.008) and 0.005 (-0.005 to 0.016, p = 0.313) across the three measurement methods. Time to complete healing at 42 days yielded Hazard Ratios (unadjusted) of

compromise patient privacy, and participants did not consent for their data to be made publicly available. We are however able to share non-identifiable participant level data upon request with an appropriate data sharing agreement in place at following email: bctudatashare@contacts.bham.ac.uk For more information, please click at the link: https://www.birmingham.ac.uk/research/bctu/data-sharing-and-protection-policy.

**Funding:** This research was funded by the National Institute for Health Research (NIHR) (NIHR200132) using UK aid from the UK Government to support global health research. RL and PG are supported by NIHR ARC West Midlands; and PG is NIHR Senior Investigator. The views expressed in this publication are those of the author(s) and not necessarily those of the NIHR or the UK Department of Health and Social Care. The funders had no role in study design, data collection and analysis, decision to publish or preparation of the manuscript. All authors received salary from the funder- NIHR, UK.

**Competing interests:** The authors have declared that no competing interests exist.

1.3 (0.8 to 2.1, p = 0.300) assessed by unblinded local clinicians and 1.2 (0.7 to 2.0, p = 0.462) on blind assessment.

## Conclusion

Any benefit from L-PRF appears insufficient to justify routine use in care of neuropathic ulcers in leprosy.

## Trial registration

ISRCTN14933421. Date of trial registration: 16 June 2020.

## Author summary

Autologous Blood Products such as Leucocyte and Platelets Rich Fibrin (L-PRF) gel have been widely used in the treatment of skin ulcers, including Leprosy ulcers. L-PRF gel contains a high concentration of platelets and leucocytes rich in growth factors that might promote wound healing. However, our systematic review showed that there is a lack of high-quality trial evidence for the effectiveness of L-PRF. To evaluate the efficacy of L-PRF we conducted an individual randomised controlled trial (n = 130) of L-PRF gel vs conventional normal saline dressings. Photographs were taken during twice weekly dressing changes in both intervention and control groups. The ulcer photographs were assessed by blinded assessors using three different ulcer area measurement methods. Outcomes were rate of healing and time to complete epithelialisation. Point estimates for intervention effects across outcomes and measurement methods, intervention vs control was favorable to L-PRF but the effect sizes were small and mostly not statistically significant.

## Introduction

Autologous blood product extracts have been widely used to promote healing in many parts of the body. Blood products are classified into four categories based on the leukocyte and fibrin, namely Pure Platelet-Rich Plasma (PRP), Leukocyte and Platelet Rich Plasma (LPRP), Pure Platelet Rich Fibrin (P-PRF) and Leukocyte and Platelet Rich Fibrin (LPRF) [1]. The two most widely used products are PRP and L-PRF. They release a variety of concentrated growth factors to promote healing, including platelet-derived growth factor, transforming growth factor, vascular endothelial growth factor, epidermal growth factor, fibrinogen and insulin like growth factor [2].When applied to the surface of a skin ulcer their healing properties have been compared to those of the scab that forms on the surface of a traumatic ulcer [3]. These products have also been used to promote healing in other parts of the body, and have been evaluated in randomised trials on tendon injury [4–8] and to promote healing of sockets following tooth extraction [9,10]. In addition, these preparations have been widely used in diabetic foot ulcer treatment [11–18] Furthermore, these have been also used in treatment of venous ulcers [19,20] However, our systematic review on surface application of autologous blood products in skin ulcers found only seven RCTs, only one of which had low risk of bias; most were small trials (15 to 120 patients), and results were heterogeneous [21]. Five of the 7 RCTs included in the systematic review concerned diabetic ulcers [22–26] and remaining two concerned venous ulcers [19,20].

Given the wide and increasing use of autologous blood product therapy in ulcers generally, and in Leprosy specifically we carried out TABLE (a Trial of Autologous Blood products to promote ulcer healing in Leprosy). This trial aimed to evaluate the effectiveness of autologous blood products on wound healing in leprosy neuropathic foot ulcers. Platelet Rich blood products in the forms of L-PRF or PRP have been recently widely used in the treatment of chronic neuropathic ulcers in Leprosy [27]. There were only two presentations about the L-PRF/PRP in wound healing in International Leprosy Congress (ILC)-2019 in Manila -one about the application of L-PRF in leprosy foot ulcers [28] and another about the role of PRP in diabetic and leprosy foot ulcers [29], whereas there were 8 papers (7 oral papers and one e-poster paper) about the use of L-PRF/PRP in leprosy foot ulcers presented at ILC-2022 in Hyderabad, India [30]. We used L-PRF gel that is rich in factors including platelet-derived growth factors (PDGF), vascular endothelial growth factors (VEGF) and epidermal growth factors (EGF).

## Methods

### Ethics statement

This study was approved in the UK by the University of Birmingham Biomedical and Scientific Research Ethics Committee (BSREC ERN 19–1960) and locally in Nepal through the Nepal Health and Research Council (NHRC ERB 303/2020 P).

### Study design

TABLE was a two-arm parallel individually randomised controlled trial with blinded outcome assessment conducted at the Anandaban Hospital, The Leprosy Mission Nepal in Kathmandu. The protocol for the trial has been published [31].

### Participants

The population from which eligible patients were drawn comprised of patients referred from all parts of Nepal to the Anandaban Hospital for the treatment of chronic neuropathic ulcers of the foot due to leprosy. Patients were eligible if they were an adult ($\geq$18 years of age) with a foot ulcer of at least six weeks duration, the ulcer was not infected (without any pathogenic organism growth in ulcer swab culture) and had an area of 2 to 20 cm$^2$. We excluded patients with anaemia (haemoglobin less than 9 gm/dL), auto-immune leprosy reactions, requiring surgical care such as skin-graft, uncontrolled hypertension or other serious medical conditions (e.g. diabetes or diabetic ulcer, HIV, chronic Hep B and TB patients under active treatment). People with large ulcers and people with anaemia were excluded because of the need to perform regular venesection to prepare L-PRF.

Eligible patients were offered entry in the trial at the point where their clinician judged them suitable for treatment i.e. when the ulcer was clear of debris or infection. They were invited to participate by a research fellow trained to Good Clinical Practice standards. A patient information leaflet was provided and the potential participant was invited to sign a consent form on the following day. Written consent was obtained from all participants. Patient information leaflet and consent forms were translated from English and back-translated according to the WHO methodology for high fidelity translation [32].

### Randomisation

Following consent and baseline data collection, individual participants were randomised in a 1:1 ratio to receive L-PRF vs. control treatment (normal saline dressing). We used a permuted block randomisation with randomly selected block sizes of 2, 4, 6, or 8 to maintain balance between

numbers allocated to each group. The randomisation sequence was generated by the trial statistician at the NIHR registered Birmingham Clinical Trials Unit, University of Birmingham using a random number generator. The table generated by this method was uploaded to the Research Electronic Data Capture (REDCap) software used to enrol participants and was inaccessible to members of the trial team at the study site to ensure allocation concealment. The software returned a unique and inviolable study number, when the patient was entered in the TABLE trial.

## Intervention, control and activity monitoring

**Intervention.** The intervention involved application of Leukocyte and Platelet-Rich Fibrin (L-PRF) matrices, prepared using the participant's own blood, and applied to the ulcer bed. This method has been in routine use at Anandaban Hospital for many years and has been described in detail [31]. In brief, up to 80 ml of the participant's blood (depending on the size of ulcer) was collected twice per week at the time of routine dressing change carried out in a minor operating room. The blood was then centrifuged to obtain the fibrin matrix gel, which was compressed and applied to the ulcer bed and then covered with a Vaseline gauze dressing.

**Control.** Participants in the control group received usual care of twice-weekly normal saline and a Vaseline gauze dressing and did not have any blood taken. The clinical care of these participants was identical to the intervention participants. Ward staff were not aware of the participant's treatment, but patients and trial team in the hospital were not blinded.

All participants (including those in the control group) were given iron, folic acid, multivitamins and vitamin-C tablets. Both control and intervention groups received routine twice-weekly dressing changes by trained nurses or paramedics until their ulcers healed (complete re-epithelialisation), or up to a maximum of 70 days. Any missed dressing change sessions were noted but not treated as a deviation from the protocol.

In the event that a participant had more than one ulcer, the largest ulcer was selected as the index ulcer for analysis purposes before randomisation.

**Activity measurement.** Since patients and some staff were not blinded, we took measures to detect potential 'performance bias', whereby patients in one of the groups might alter their ambulatory behaviour and thereby affect objective healing rates. Participants (intervention and control) were therefore asked to wear a pedometer (Model: Mi Smart Band 5, Model: XMSH10HM) on the ankle of their non-affected limb from the first dressing change until 42 days or discharge, whichever came first. This enabled us to monitor activity across intervention and control groups and thereby detect any reduced use of the legs in the intervention group. We monitored consecutive participants, starting at the 42$^{nd}$ participant, which is the point where the Chair of the Steering Group alerted us to the issue.

## Outcome assessments

**Cross-over at 42 days.** Clinicians involved in the trial felt that patients should be offered L-PRF at their discretion if the patient's ulcer was healing slowly. It was decided, therefore, and approved by the ethics committee, that patients in the control group should be able to 'cross-over' to receive L-PRF after 42 days if their ulcer was healing slowly. For this reason, outcomes at 42 days are the main outcomes in the trial. At this point it was predicted that about 70% of ulcers would have healed [31].

**Main outcomes.**

1. "Rate of healing" (cm$^2$ per day) based on two observations per week of ulcer size censored at 42 days. We estimated the effect of the intervention on the change in the ulcer size per unit time, i.e. the rate of healing (see Statistical Analysis).

2. Time to complete re-epithelialisation (censored at 42 days).

Both endpoints were analysed with and without adjustment for baseline characteristics (trial baseline ulcer area measured using the PUSH tool and participants' age for the time to complete re-epithelialisation outcome and only participants' age for the rate of healing outcome since the rate of healing was measured from the baseline time point).

**Other outcomes.**

1. Rate of healing based on two observations per week up to 70 days ($cm^2$ per unit time).

2. Time to complete re-epithelialisation (observed up to 70 days).

3. Generic Quality of Life (QoL) measured at first dressing change, and then fortnightly during in-patient stay and at six-month follow-up, using the EQ-5D 3L. This scale has previously been used and validated in a Nepali population [33].

**Long-term outcomes.** At 6-month follow-up from randomisation we observed the proportion of participants with:

1. Recurrence of treated ulcer.

2. Appearance of new ulcer.

**Additional resource use outcomes.**

1. Days hospitalised prior to discharge and total (to include any readmission due to leprosy/ulcers) by 6 months.

2. Duration of dressing changes.

All secondary outcomes were analysed by adjusting for baseline characteristics (trial baseline ulcer area measured using PUSH tool and participants' age) unless otherwise stated.

## Data collection

**Sequencing of data collection.** Demographic data (age and sex), clinical data (number and size of ulcers), and concurrent diseases was collected for all consenting participants before randomisation. Data on quality of life (QoL) information (measured using the EQ-5D-3L) was collected fortnightly from first dressing change until discharge. Ulcer measurements are discussed below. Steps taken were collected daily (from pedometer readings) until 42 days (the point where cross-over may occur) or discharge, whichever came first. Follow up data collection at six months was observed during an outpatient appointment or a home visit. Data, including photographs of ulcers, were collected on electronic tablets using the REDCap system by local researchers. There was thus a 'closed loop' between ulcer image capture in Kathmandu and images were sent to Birmingham for blinded measurement of ulcer area and complete healing.

## Procedure of intervention and control

A videotape of the procedure can be found here (https://www.youtube.com/playlist?list=PLj7Tja5A6syiB3A6SsjhudFfewpyR7usM).

**Ulcer measurements.** *Photographs.* Standardised photographs [34] were taken twice weekly during dressing changes and at the six month follow up visit in both intervention and control groups.

*Optical systems and sequential measurements.* Photographs were obtained using two different optical systems. The first optical system was the ARANZ SilhouetteStar device (camera)

where the image was synchronised to the participant's trial number and saved to the secure ARANZ SilhouetteCentral SQL server. The ARANZ camera has an inbuilt mechanism that ensures that each image is taken at the optimal distance from the image and uses this distance in its calculations. The image was transferred to the secure server. The tool was then used to measure the ulcer by two methods. The first method used manual tracing whereby the ulcer boundary was 'hand' drawn in the ARANZ software ("ARANZ Manual"). The software was then used to calculate the ulcer dimensions based on this outline. The second method used an automatic tracing feature in the software ("ARANZ Auto"). In this instance, the blind assessor delineates a region of interest (extending beyond the boundary of the ulcer) and the validated software automatically locates the ulcer boundary and calculates the wound dimensions.

The second optical system used the in-built camera in the tablet devices (Samsung Galaxy Tab S6) used to collect all trial data. The photograph was taken perpendicular to the ulcer. For calibration purposes, a 3 cm size clean paper ruler with date and participant's trial identification number was placed in the photograph frame above or below the ulcer but at the level of the skin. The photograph was uploaded to the server at the University of Birmingham and evaluated digitally in Birmingham using the PictZar Digital Planimetry Software [35] with an electronic PUSH Tool (National Pressure Injury Advisory Panel (NPIAP) at https://npiap.com/page/PUSHTool) ("PUSH Manual"). The blind assessor delineated an area of interest by manually 'painting' the ulcer area with colour using a computer mouse. The software then calculated the ulcer dimensions based on this profile. Each ulcer was thus measured using two optical systems yielding three measurements in total.

*Complete epithelialisation*. The date at which complete re-epithelialisation took place was recorded in two ways. First, it was determined by the local clinician, and recorded on the tablet. Second, the blinded assessors independently determined the point of complete re-epithelialisation based on an assessment across all three measurement methods (two with the ARANZ system, one with the PUSH Tool).

*Assessments*. All photographs were transferred to the University of Birmingham for measurement and to determine the point of complete re-epithelialisation. Blind assessors were trained on all techniques of measurement, as well as the use of the software, by the software developers. For quality assurance, five measured images at the outset were sent to the companies (PictZar and ARANZ Medical) for approval. Once the ulcer was completely re-epithelialised or 70 days had elapsed from randomisation, whichever occurred first, the ulcer images were assembled. The two images from a given patient were randomly allocated to at least one blind assessor. The blind assessor then made the above three measurements. In addition, a randomly selected 20% of the image pairs from one patient at one dressing change, were assigned to both blind assessors to test inter-rater reliability.

## Sample size

The sample size was based on time to complete re-epithelialisation. As stated, we assumed that 70% of ulcers would heal within 42 days with standard care [36]. Further, assuming that the intervention would increase this proportion to 90% and hazards are constant and proportional, for a two-sided test of the hazard ratio with a type 1 error of 5% and statistical power of 80% and a 1:1 allocation ratio, 47 individuals were required in each group. To allow for withdrawals and right censoring, we aimed to recruit 65 patients in each group. We expected rate of healing (our other main outcome) to provide more precise estimates, since it involved multiple observations of a continuous variable.

### Analyses and inference

**Statistical methods for main endpoints.**   Time to healing was analysed using a Cox proportional hazards model with and without adjustment for baseline characteristics (trial ulcer area and participants' age) allowing for right-censoring. For the rate of healing analysis, we defined the outcome as the ulcer size in $cm^2$ at each time point and included in the model a function of time since randomisation, treatment status and their interaction. We compared different functions including quadratic, linear, and log. Our parameter(s) of interest were the interaction term(s). We derived the mean difference in rate of healing (in $cm^2$ per unit time) between intervention and control groups from this model. The model also included participant-level random effects both with and without adjustment for participant characteristics. Given we fit the model with data from three different measurement procedures and with and without adjustment, we adjusted reported for multiple testing using the Hochberg stepdown method, which provides an efficient means of controlling the family-wise error rate [37]. However, our protocol states, "we will not (miss) use statistical tests as decision rules" and leave "the reader to make an interpretation". In line with the American Statistical Association Statement [38] we report point estimates and confidence intervals and p-values without dichotomising results into "significant" and "not significant".

We assessed agreement between the blind assessors for each of the three measurement methods (ARANZ Auto, ARANZ Manual, and PUSH Manual) using Bland-Altman plots. However, given the randomisation of images to blind assessors, we would not expect lack of agreement to cause bias.

**Methods for additional analyses.**   We pre-specified one sub-group: ulcer size at baseline (measured using the PUSH tool). Ulcer size at baseline was dichotomised (below vs. at or above the median value). The sub-group analysis was conducted with and without including participants' age.

Quality of life (QoL) was analysed by calculating and comparing area under the curve (AUC) across

across intervention and controls. Baseline results were triangulated with clinical observations to avoid bias and to determine how differences in healing rates correspond to differences in quality-of-life end-points. A full economic analysis will be reported separately.

Average daily step count between treatment and control groups was compared as a simple difference in means (t test). Since one group may have stayed longer in hospital than the other and since there may have been an interaction between rate of healing and step count, we compared step counts over periods pre-set at 7, 14 and 42 days.

**Reporting.**   Where a participant withdrew from the intervention and further data collection, no further data was collected. Where the participant withdrew from the randomised intervention but agreed to contribute data, they were followed-up to the point of discharge as an inpatient. The trial is reported in line with the CONSORT (Consolidated Standards of Reporting Trials) Standards [39,40].

### Patient and public involvement

People affected by leprosy were involved in the design of the study protocol and also involved during the trial. The Trial Management Group and Trial Steering Committee also included people affected by leprosy.

## Results

Participant enrolment took place from 18th September 2020 to 24th May 2022, with follow-up completed on 23rd November 2022. Of 283 patients screened for eligibility, 132 were consented

and eligible for randomisation and 130 were randomised (S1 Table). The full list of reasons for participant exclusion is given in (S2 Table).

A quantitative description of the number of dressing changes, time for dressing changes, volume of blood drawn and number of L-PRF application are described in (S3 Table).

There were 10 protocol deviations and 30 file-notes during the trial, details of protocol deviations can be found in (S4 Table).

### Baseline characteristics

Baseline characteristics were well matched (Table 1); the mean age of participants was 54.0 years, and 27 (20.8%) were female. Four participants withdrew during the trial, two before, one on and one after the 42 days milestone. All participants contributed to the analyses up to the time point they were followed up.

There were no cross-overs before 42 days but two participants missed a dressing change on at least one occasion. Full details can be found in S6 Table. Seven participants in the control crossed over to the intervention group after the 42 days cut-off.

### Outcomes at 42 days

**Healing rates.** The best fitting healing rate model included a quadratic function of time. The mean differences in daily healing rates over the first 42 days from randomisation are given in Table 2. Adjusted and unadjusted mean differences in daily healing rates are given for all three measurement methods. The point estimates favour the intervention, but the effect sizes are small. For example, with the ARANZ auto-tool (which provides the least inter-observer variation- see below) the mean difference in daily healing rate was 0.012 cm$^2$/day (95% CI: 0.001 to 0.023), compared to a baseline mean ulcer area of 3.8 (SD: 2.9) cm$^2$ used when adjusting the models. Confidence intervals ranged from 0.004 to 0.027 cm$^2$/day with the ARANZ manual tool to -0.005 to 0.016 cm$^2$/day with the PUSH tool.

Fig 1 shows the estimated healing rate functions in both absolute and proportional (to the baseline ulcer area) terms for the three measurement methods.

**Complete epithelialisation.** We estimated the hazard ratio comparing time to complete re-epithelialisation between treatment and control. Participant follow-up times were censored at 42 days from randomisation. As shown in Table 3, hazard rates are somewhat in favour of the intervention in both the local clinical and blinded assessments of complete healing and in adjusted and unadjusted models. However, the data are compatible with effects in either direction. Time to complete healing at 42 days, for both clinical assessment and blind assessment, is also shown on Kaplan-Meier plots in Fig 2.

As shown in the Table 4 time to complete re-epithelialisation outcome censored at 70 days Hazard ratios in both models (adjusted and unadjusted) favours the intervention group. However, there is differences in the healing rates assessed by local clinicians and blinded assessor.

For reasons explained in the discussion and protocol, the above observations could be biased by informative discharge by the unblinded clinician. We therefore compared proportions healed at 42 days according to whether they were assessed by clinician or blind assessor. Consistent with a small bias in favour of informative censoring, healing rates in the L-PRF group were respectively 55.4% vs. 50.8% (representing a difference of three patients) (Table 3).

Further analysis, where the p-values are adjusted for multiple testing, are shown in S7 Table.

**Sensitivity and subgroup analyses.** Sensitivity analyses based on per protocol analyses i.e. excluding the two intervention patients. Consistent with the low rate of non-adherence this does not affect the conclusions, for either rate of healing (S8 Table) or time to re-epithelialisation (S9 Table).

**Table 1. Baseline characteristics by group.**

| | | Dressing changes with normal saline (N = 65) | Dressing changes with L-PRF matrix (N = 65) | Total (N = 130) |
|---|---|---|---|---|
| **Variables used in covariate adjustment** | | | | |
| Trial ulcer Area (cm$^2$) -PUSH tool | n | 65 | 64 | 129 |
| | Mean (SD) | 3.9 (3.0) | 3.7 (2.8) | 3.8 (2.9) |
| | Min—Max | 0.6–12.5 | 0.9–11.3 | 0.6–12.5 |
| | Missing | 0 | 1 | 1 |
| Age at randomisation (years) | n | 65 | 65 | 130 |
| | Mean (SD) | 53.6 (17.1) | 54.5 (14.4) | 54.0 (15.8) |
| | Min—Max | 20.0–89.0 | 22.0–84.0 | 20.0–89.0 |
| **Participant demographics** | | | | |
| Gender | Male | 54 (83.1%) | 49 (75.4%) | 103 (79.2%) |
| | Female | 11 (16.9%) | 16 (24.6%) | 27 (20.8%) |
| | Missing | 0 | 0 | 0 |
| Highest level of education | Never joined formal school | 25 (38.4%) | 26 (40.0%) | 51 (39.2%) |
| | Can read and write | 16 (24.6%) | 7 (10.8%) | 23 (17.7%) |
| | Primary level | 13 (20.0%) | 14 (21.5%) | 27 (20.8%) |
| | Secondary level | 9 (13.9%) | 16 (24.6%) | 25 (19.2%) |
| | Higher secondary level | 2 (3.1%) | 2 (3.1%) | 4 (3.1%) |
| | University level | 0 (0%) | 0 (0%) | 0 (0%) |
| | Missing | 0 | 0 | 0 |
| **Clinical information** | | | | |
| BMI | n | 65 | 65 | 130 |
| | Mean (SD) | 22.0 (3.3) | 23.0 (3.7) | 22.5 (3.5) |
| | Min—Max | 17.0–32.5 | 14.9–32.5 | 14.9–32.5 |
| Blood Pressure–Systolic (mmHg) | n | 65 | 65 | 130 |
| | Mean (SD) | 112.8 (9.9) | 114.2 (11.8) | 113.5 (10.9) |
| | Min—Max | 90.0–140.0 | 90.0–150.0 | 90.0–150.0 |
| **Leprosy details** | | | | |
| Number of years since leprosy diagnosis | n | 65 | 65 | 130 |
| | Mean (SD) | 20.8 (15.3) | 18.6 (14.2) | 19.7 (14.7) |
| | Min—Max | 1.0–60.0 | 1.0–50.0 | 1.0–60.0 |
| **Voluntary Muscle Testing/Sensory testing (VMT/ST)** | | | | |
| VMT/ST | Normal | 0 (0%) | 0 (0%) | 0 (0%) |
| | Impaired | 65 (100%) | 65 (100%) | 130 (100%) |
| **Current Ulcer Information** | | | | |
| Number of weeks trial ulcer unhealed | n | 65 | 65 | 130 |
| | Median | 22.0 | 26.0 | 26.0 |
| | P$_{25}$—P$_{75}$ | 13.0–44.0 | 13.0–52.0 | 13.0–52.0 |
| Is the trial ulcer recurrent? | Yes | 46 (70.8%) | 35 (53.9%) | 81 (62.3%) |
| | No | 19 (29.2%) | 30 (46.1%) | 49 (37.7%) |

Data are either mean (SD) or number (%).

This table is abridged. For full table, please see in S5 Table.

Further sensitivity analyses concerning the impact of missing data are presented in S10 and S11 Tables. Unsurprisingly, given the small number of missing data, the conclusions are qualitatively similar.

**Table 2. Mean difference in daily healing rate over 42 days for each model-Primary ITT analysis.**

| | Unadjusted Model[1,2] | Adjusted Model[3,2] |
|---|---|---|
| | Mean difference (95% CI[4]) p-value | Mean difference (95% CI[4]) p-value |
| ARANZ auto tool, cm$^2$ | 0.012 (0.001 to 0.023) p = 0.027 | 0.012 (0.001 to 0.023) p = 0.028 |
| ARANZ manual tool, cm$^2$ | 0.016 (0.004 to 0.027) p = 0.008 | 0.015 (0.004 to 0.027) p = 0.009 |
| PUSH tool, cm$^2$ | 0.005 (-0.005 to 0.016) p = 0.313 | 0.005 (-0.005 to 0.016) p = 0.323 |

1: Unadjusted mixed effects regression model with time modelled as a quadratic function. Model includes interaction terms between time and treatment and time^2 and treatment.
2: Estimated average difference>0 indicates a higher daily healing rate in dressing change with L-PRF matrix group.
3: Mixed effects regression model adjusted for the baseline value of participant age, with time modelled as a quadratic function. Model includes the interaction terms between time and treatment, and time^2 and treatment. Baseline participant age was treated as a continuous variable and considered as a fixed effect in this adjustment.
4: Confidence intervals are estimated using Wald statistics.

We carried out subgroup analysis by area of ulcer size at baseline for rate of healing and complete re-epithelialisation (S12 and S13 Tables). Regarding the intervention effect, the absolute mean daily difference in healing rate was greater for large than for small ulcers but for time to complete epithelialisation, there is not much evidence to indicate that was different by baseline ulcer size groups.

## Outcomes at 70 days

As stated, seven control participants crossed over to receive intervention after 42 days. The rate of healing up to 70 days by the randomised treatment allocation and time to complete re-epithelialisation are given in (S14 and S15 Tables).

## Quality of life and resource use

The quality-of-life data until 70 days post randomisation or discharge from treatment, whichever came first, is shown in (S16 Table). The mean quality of life area under the curve was better in the L-PRF group but by a very small amount (0.04) and there is considerable uncertainty about the magnitude, and direction, of the effect (-0.02 to 0.09).

Participants in the intervention group had a shorter average length of stay in hospital by 6 days (-11.9 to -0.08). The total days hospitalised over a period 6 months from randomisation was reduced by 13.7 days in the intervention group (-26.7 to -0.7), compared to the control.

We found the L-PRF method to be more time consuming, with a difference in average dressing duration at day 0 of 17.4 minutes (16.2 to 18.5), reducing to, by an average of 9.74 seconds per day (-11.50 to -7.97), to 10.6 minutes (9.4 to 11.7) at day 42 (S1 Fig).

## Long-term outcomes

There was a high rate of ulcer recurrence observed during the six-month follow up; 22 (37%) in the normal saline group and 18 (30%) in the intervention group with little certainty about direction of effect (adjusted relative risk: 0.80 (0.49 to 1.30)). Also, there were six new ulcers in the normal saline group versus eight in the L-PRF group (adjusted relative risk: 1.27 (0.48 to 3.40)), but the confidence intervals were again wide (S17 Table).

## Analysis of activity data

Activity measurements were obtained for 44 in control group and 45 in the intervention group. The difference in number of steps taken over 42 days was 173.1 (-302.1 to 648.2),

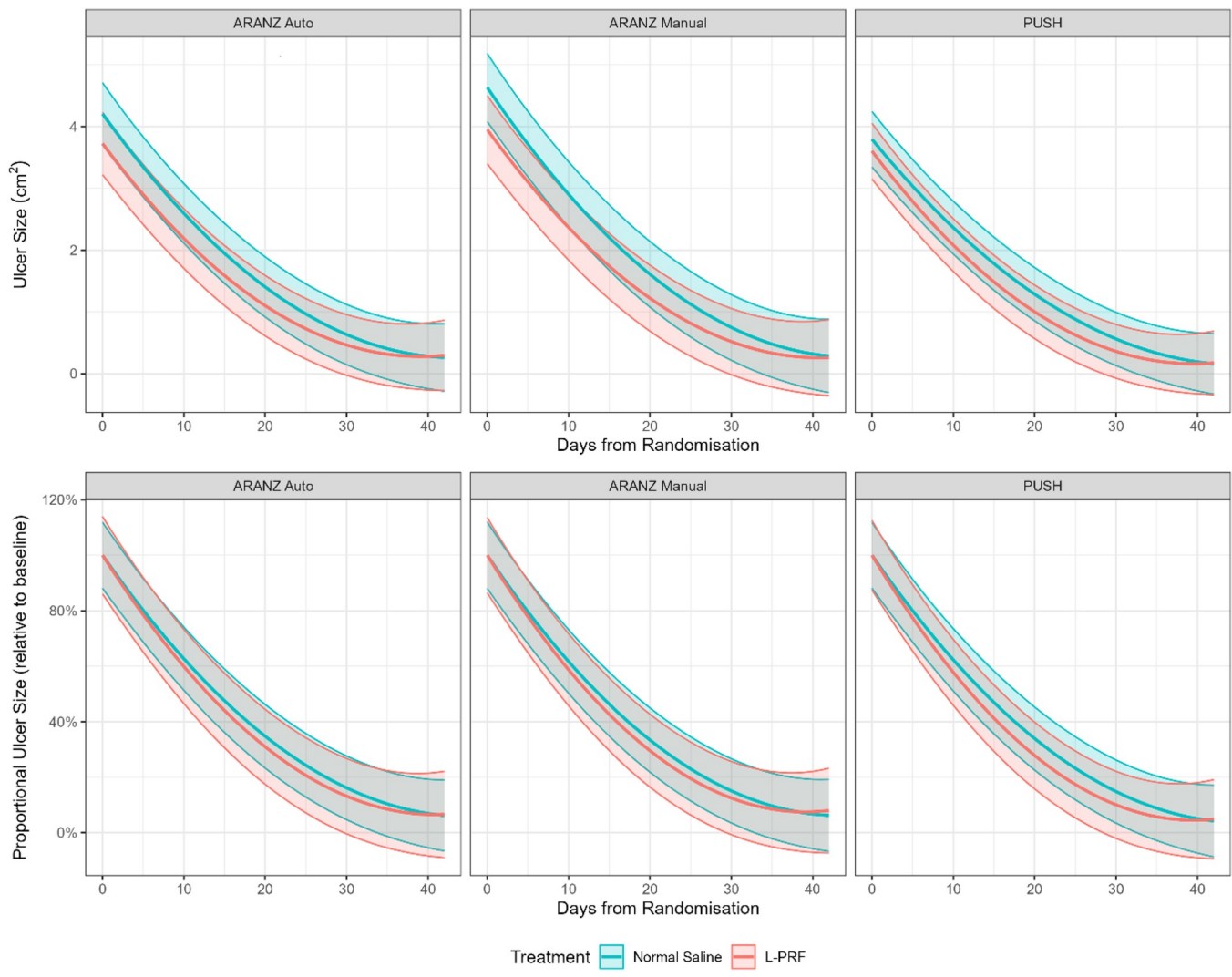

**Fig 1. Rate of healing across the three different measurement methods.**

meaning there were slightly more steps taken in the control group but with considerable uncertainty about the duration of effect (S18 Table).

## Measurement characteristics

The Bland-Altman plots assessing inter-observer agreement are shown for the three measurement methods in S2 Fig. The smallest differences between blind assessors were exhibited by the ARANZ Auto method, followed by ARANZ Manual, then PUSH Manual. ARANZ Auto also had a mean difference of approximation zero, whereas the other two methods had a small systematic difference in mean area measure between the blind assessors. Finally, we did not find any evidence of proportionate bias in agreement.

## Discussion

### Principal findings

Prior to this study there was considerable enthusiasm for L–PRF ulcer therapy but little high-quality evidence and hence scientific uncertainty [41–43]. However, the effect sizes observed

**Table 3. Time to complete re-epithelialisation analysis censored at 42 days.**

| | Dressing changes with normal saline (n = 65) | Dressing changes with L-PRF matrix (n = 65) | Unadjusted Hazard Ratio[2] (95% CI) p-value | Adjusted[1] Hazard Ratio[2] (95% CI) p-value |
|---|---|---|---|---|
| **Healing assessed by clinician** | | | | |
| Number of censored participants[3] | 33 (50.8%) | 29 (44.6%) | 1.3 (0.8 to 2.1) p = 0.300 | 1.4 (0.8 to 2.2) p = 0.206 |
| Number of healed participants | 32 (49.2%) | 36 (55.4%) | | |
| **Healing assessed by blinded assessor[4]** | | | | |
| Number of censored participants[3] | 33 (50.8%) | 32 (49.2%) | 1.2 (0.7 to 2.0) p = 0.462 | 1.3 (0.8 to 2.0) p = 0.370 |
| Number of healed participants | 32 (49.2%) | 33 (50.8%) | | |

1: Cox proportional hazards adjusted for the baseline values of trial ulcer size and participant age. Trial ulcer size and participants' age were both treated as continuous variables and considered as fixed effects in this adjustment.

2: HR>1 means–Participants in Dressing Changes with L-PRF Matrix Group are more likely to have completely re-epithelialised ulcers than participants in Dressing Changes with Normal Saline Group.

3: One participant in the dressing changes with normal saline group withdrew before having reached 42 days post-randomisation.

4: There were some cases in which the blinded assessor classed the ulcer as healed in one image, as unhealed in the next image, and healed in the subsequent image. For these cases, we defined the time to healing as the time to the first instance in which the blinded assessor diagnosed complete healing.

in this study are small. While the point estimates generally favour L-PRF, the data were only compatible with small, and arguably not clinically important, differences. In the case of time to complete epithelialisation [44], the confidence intervals show considerable uncertainty about the direction of any effect. In the case of ulcer healing rates, there was a small effect in favour of the treatment. However, in follow up analyses (to be published in a forthcoming paper) we found that these results were highly sensitive to the functional form of the model (such as using a log-linear model, and more complex non-linear functions of time post-randomisation), and specification of the treatment effect (differences in areas or rates).

The intervention has costs including the need to train staff, purchase equipment and follow meticulous anti-sepsis during preparation of the blood product. We discuss the costs in more detail in a forthcoming paper, but found the method to be more time consuming taking between 17 and 11 extra minutes per dressing depending on time since randomisation up to

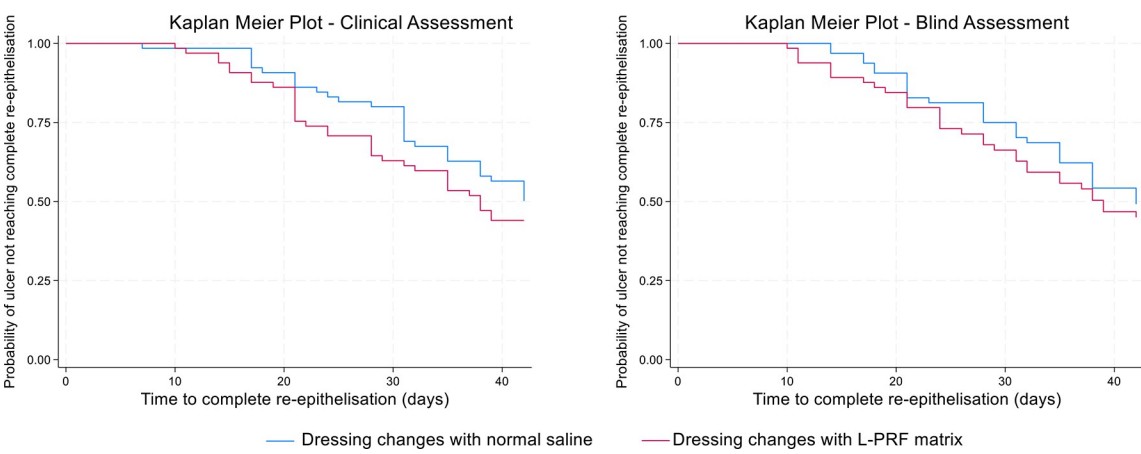

**Fig 2. Kaplan Meier estimates for the time of complete re-epithelisation at 42 days.**

**Table 4. Time to complete re-epithelialisation outcome censored at 70 days (ITT analysis).**

| | Dressing changes with normal saline (n = 65) | Dressing changes with L-PRF matrix (n = 65) | Unadjusted Hazard Ratio[2] (95% CI) p-value | Adjusted[1] Hazard Ratio[2] (95% CI) p-value |
|---|---|---|---|---|
| Healing assessed by clinician | | | | |
| Number of censored participants[3] | 12 (18.5%) | 9 (13.8%) | 1.4 (0.9 to 2.0) p = 0.118 | 1.5 (0.8 to 2.2) p = 0.044 |
| Number of healed participants | 53 (81.5%) | 56 (86.2%) | | |
| Healing assessed by blinded assessor[4] | | | | |
| Number of censored participants[3,5] | 14 (21.5%) | 15 (23.1%) | 1.2 (0.8 to 1.8) p = 0.265 | 1.3 (0.9 to 2.0) p = 0.147 |
| Number of healed participants | 51 (78.5%) | 50 (76.9%) | | |

1: HR>1 means—Participants in Dressing Changes with L-PRF Matrix Group are more likely to have completely re-epithelialised ulcers than participants in Dressing Changes with Normal Saline Group.

2: Cox proportional hazard model adjusted for the baseline values of trial ulcer size and participant age. Trial ulcer size and participant age was treated as continuous variables and considered as fixed effects in this adjustment.

3: One participant in the dressing changes with normal saline group and 3 participants in the dressing changes with L-PRF matrix group withdrew before having reached 70 days post-randomisation.

4: There were some cases in which the blinded assessor classed the ulcer as healed in one image, as unhealed in the next image, and healed in the subsequent image. For these cases, we defined the time to healing as the time to the first instance in which the blinded assessor diagnosed complete healing.

5: There was one participant whose ulcer was healed at 71 days but their time to complete re-epithelialisation was censored at 70 days.

42 days. For someone needing, say, 10 dressings this would amount to about two and a half hours. The relatively large volumes of blood add a further disadvantage to the method. While it is possible that L-PRF may have a small effect on ulcer healing, it is hard to reach a conclusion in favour of L-PRF for ulcer treatment. We leave it to others to conclude on whether neuropathic ulcers of leprosy are so different from those of, say, diabetes as to warrant a separate trial in that or any other ulcer type.

## Comparison with other studies

We have conducted a systematic review of RCTs of an autologous blood product vs. standard dressings in the treatment of non-traumatic skin ulcers [21]. Only 7 studies (two L-PRF and five PRP gel) met our inclusion criteria. The highest quality paper showed a significant difference in favour of the intervention for time to complete healing only and not their other outcome, rate of healing [45]. Meta-analysis was impossible as the data were not provided or there were zero instances of healing in the control group in time to complete healing/proportion completely healed comparisons. The unimpressive effects observed here contrast with some of the results of the reviewed studies. However, a high-quality study of L-PRF in tendon repair also found no evidence of effectiveness within narrow CIs [5].

## Limitations and strengths

Recruitment into the trial began after the first COVID-19 lockdown was eased and continued during periods of restricted travel. Nevertheless we were able to recruit to the study and carry out all planned data collection, in keeping with the schedule of dressing changes and study timelines, with a low rate of attrition.

To our knowledge TABLE is the largest trial of a blood product regenerative treatment vs. standard dressings alone for treatment of skin ulcers. In contrast to other studies, we had

allocation concealment and blinded measurements of ulcer outcomes (our study used a 'closed loop' between ulcer photographs in Nepal and blind assessors in Birmingham). Each ulcer site was assessed in multiple (three) ways and blind assessor reliability was assessed. It would have been very difficult to blind participants and clinicians since that would have required repeated venesection from control participants and elaborate procedures to obscure the preparation and application of the blood-derived product. We therefore devised a method that would provide evidence of the plausible causal mechanisms for performance bias—namely reduced ambulation among intervention vs. control participants. The activity monitoring was successfully accomplished and showed little (and non-significant) difference between groups.

Our study has a limitation in that (as mentioned in the protocol) earlier discharge in the intervention group could lead to 'informative' censoring. It would have been unethical to use an adjudication process that could delay discharge. Any bias for this reason would likely have favoured the intervention, in which case L-PRF would have had (even) less impressive effects. Consistent with this point, the effect size was (even) less impressive when we made the analysis on the basis of independent blind assessor observation of complete healing.

We did not have resources to analyse the biological characteristics of the platelet and leukocyte rich plasma. However, the technique was carried out by an experienced team using standard and recommended methods.

## Conclusion

Point estimates point to a small benefit for L-PRF that, arguably, is not sufficient in magnitude to justify the resources used. Any bias due to informative censoring would have exaggerated the observed treatment effect. Our study has also provided evidence relating to measurement of ulcer size, reliability of recording of complete epithelialisation and the performance of different models of healing rates to which ulcer area data may be fitted. A separate paper on these methodological issues will be submitted for publication imminently.

## Supporting information

**S1 Table. Consort flow diagram.**
(DOCX)

**S2 Table. Reasons for participant exclusion.**
(DOCX)

**S3 Table. Description of the intervention -up to 70 days.**
(DOCX)

**S4 Table. Protocol deviations by group.**
(DOCX)

**S5 Table. Full table of baseline characteristics by group.**
(DOCX)

**S6 Table. Adherence to allocated intervention measured at 42 days.**
(DOCX)

**S7 Table. P-Values adjustment for multiple testing (ITT analysis).**
(DOCX)

**S8 Table. Mean difference in daily healing rate over 42 days for each model—(Per-protocol (adherent) analysis).**
(DOCX)

**S9 Table. Analysis of primary outcome measure–Time to complete re-epithelisation censored at 42 days (Per-protocol (adherent) analysis).**
(DOCX)

**S10 Table. Mean difference in daily healing rate over 42 days for each model—Missing data analysis.**
(DOCX)

**S11 Table. Baseline characteristics of those who provided data that allows the rate of healing outcome using the ARANZ automated method to be assessed, vs. those who did not.**
(DOCX)

**S12 Table. Mean difference in daily healing rate over 42 days for each model–Subgroup analysis.**
(DOCX)

**S13 Table. Subgroup analysis of the time to complete re-epithelisation outcome assessed by clinician censored at 42 days and including the area of the baseline ulcer.**
(DOCX)

**S14 Table. Area of ulcer size using each of the three tools at chosen time points censored at 70 days.**
(DOCX)

**S15 Table. Analysis of the time to complete re-epithelisation outcome censored at 70 days (ITT analysis).**
(DOCX)

**S16 Table. Analysis of continuous secondary outcome- Quality of life 'area under the curve' until discharge or 70 days post randomisation, whichever occurred first.**
(DOCX)

**S17 Table. Recurrence of treated ulcer 6 months from randomisation.**
(DOCX)

**S18 Table. Analyses of activity measurement at 7-, 14-, and 42-days post-randomisation.**
(DOCX)

**S1 Fig. Average duration of a dressing change over time for each group.**
(DOCX)

**S2 Fig. Bland-Altman plots assessing inter-observer agreement for the three measurement methods.**
(DOCX)

## Acknowledgments

We thank the contribution of patients and trial team members at the recruitment site who contributed to data collection and data entry and members of the data monitoring and ethics committee: Ewen M Harrison, Sarah Brown, Krishna Prasad Dhakal and Namita Ghimire. We also thank the Birmingham Clinical Trials Unit and the trial steering committee for its advice and support during the study: Gemma Slinn, Premal Das, Kara Hanson, Paul Saunderson, Magdalena Skrybant and our community engagement contributors from IDEA Nepal that were also members of the Trial Steering Committee, Amar Timilsina and Dinesh Basnet. We want to

thank Bahadir Celiktemur and Tom Lingard, who performed blinded assessments of all photographs of ulcers. We thank Paul Saunderson for pointing out the risk, albeit likely small of performance bias that led us to the idea of movement monitoring. We want to express our cordial thanks to the whole clinical team at Anandaban Hospital, especially Mr Ram Kumar Maharjan, Mrs Dipawali Rana, Mrs Sabina Shrestha, Mrs Laxmi Tamang and Kashi Nath Aryal. We appreciate their dedication in the treatment of patients during the trial, especially during the COVID-19 pandemic.

## Author Contributions

**Conceptualization:** Indra B. Napit, Dilip Shrestha, Paramjit Gill, Jo Sartori, Samuel I. Watson, Richard Lilford.

**Data curation:** Eleni Gkini, Jon Bishop, Samuel I. Watson.

**Formal analysis:** Eleni Gkini, Jon Bishop, Samuel I. Watson.

**Funding acquisition:** Paramjit Gill, Jo Sartori, Samuel I. Watson, Richard Lilford.

**Investigation:** Indra B. Napit, Dilip Shrestha, Karuna Neupane, Anju Adhikari.

**Methodology:** Indra B. Napit, Dilip Shrestha, Paramjit Gill, Samuel I. Watson, Richard Lilford.

**Project administration:** Indra B. Napit, Dilip Shrestha, Sopna Choudhury, Jo Sartori, Richard Lilford.

**Resources:** Indra B. Napit, Sopna Choudhury, Jo Sartori, Richard Lilford.

**Software:** Eleni Gkini, Jon Bishop, Samuel I. Watson.

**Supervision:** Indra B. Napit, Dilip Shrestha, Sopna Choudhury, Jo Sartori, Samuel I. Watson, Richard Lilford.

**Validation:** Indra B. Napit, Sopna Choudhury, Eleni Gkini, Onaedo Ilozumba, Paramjit Gill, Jon Bishop, Jo Sartori, Samuel I. Watson, Richard Lilford.

**Visualization:** Eleni Gkini, Jon Bishop, Samuel I. Watson, Richard Lilford.

**Writing – original draft:** Indra B. Napit, Dilip Shrestha, Sopna Choudhury, Eleni Gkini, Onaedo Ilozumba, Paramjit Gill, Karuna Neupane, Anju Adhikari, Richard Lilford.

**Writing – review & editing:** Indra B. Napit, Dilip Shrestha, Sopna Choudhury, Eleni Gkini, Onaedo Ilozumba, Paramjit Gill, Karuna Neupane, Anju Adhikari, Richard Lilford.

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
