## [Decision Letter · Decision Letter 0]

10 Jan 2024

Dear Dr Napit,

Thank you very much for submitting your manuscript "A randomised Trial of Autologous Blood products, leukocyte and platelet-rich fibrin (L-PRF), to promote ulcer healing in LEprosy: the TABLE trial." for consideration at PLOS Neglected Tropical Diseases. As with all papers reviewed by the journal, your manuscript was reviewed by members of the editorial board and by several independent reviewers. In light of the reviews (below this email), we would like to invite the resubmission of a significantly-revised version that takes into account the reviewers' comments. 

We cannot make any decision about publication until we have seen the revised manuscript and your response to the reviewers' comments. Your revised manuscript is also likely to be sent to reviewers for further evaluation.

Sincerely,

Linda B Adams

Academic Editor

Mathieu Picardeau

Section Editor

Reviewer's Responses to Questions

**Key Review Criteria Required for Acceptance?**

**Methods**

-Are the objectives of the study clearly articulated with a clear testable hypothesis stated?

-Is the study design appropriate to address the stated objectives?

-Is the population clearly described and appropriate for the hypothesis being tested?

-Is the sample size sufficient to ensure adequate power to address the hypothesis being tested?

-Were correct statistical analysis used to support conclusions?

-Are there concerns about ethical or regulatory requirements being met?

Reviewer #1: The study design has a major flaw in that the primary cause of neuropathic foot ulcers is pressure. These ulcers will heal with pressure relief using a variety of dressings as long as good wound care principles are maintained. The study only references use of a Pedometer to account for walking distances but no mention of offloading devices to reduce targeted pressure at the ulcer site. Foot anatomy and gait mechanics are crucial factors in the formation of a neuropathic ulcer. These variables are not addressed in the study. Until a wound is appropriately offloaded, and the pressure removed, comparison of wound healing strategies cannot be adequately compared.

Reviewer #2: -Are the objectives of the study clearly articulated with a clear testable hypothesis stated?

Lines 78-81: The authors provide the title of the trial which one can speculate its objective, but I feel that the objective of the study is not explicitly described.

-Is the study design appropriate to address the stated objectives?

Yes.

-Is the population clearly described and appropriate for the hypothesis being tested?

Yes.

-Is the sample size sufficient to ensure adequate power to address the hypothesis being tested?

Yes. The study might have yielded some significant results with more samples, it is a challenging study to have many patients. 130 patients is a good number of this study.

-Were correct statistical analysis used to support conclusions?

Yes.

-Are there concerns about ethical or regulatory requirements being met?

No.

Reviewer #3: The methodology is well described. However, we suggest that it be clear to analyze the variables at the end of 42 days and 70 days. This way the discussion also became individualized. It is clear that the methodology used for the variables after 42 days was not rigorously presented in the manuscript. Evaluate whether the results after 42 days could result in another manuscript. We also suggest that it is possible to make the monetary value of the procedures performed in the acquisition of blood products more explicit to compare with the control group. This would allow for more robust cost analysis.

**Results**

-Does the analysis presented match the analysis plan?

-Are the results clearly and completely presented?

-Are the figures (Tables, Images) of sufficient quality for clarity?

Reviewer #1: (No Response)

Reviewer #2: -Does the analysis presented match the analysis plan?

Yes.

-Are the results clearly and completely presented?

Yes.

-Are the figures (Tables, Images) of sufficient quality for clarity?

Yes.

Reviewer #3: 1-We verified that the item "Recruitment" would be in the Methodology section and not in results.

2- "27 participants were female" we suggest also putting it as a percentage, this way we will have a better view of the percentage

3- the participant who gave up after 42 days should not be classified as a dropout, since the main results of the work ended after 42 days (70%)

4- It would be interesting if the manuscript time limit was until the end of 70 days with the inclusion of all variables as reliable findings.

5- See the possibility of combining the information in a table with just the data from 42 days to 70 days.

**Conclusions**

-Are the conclusions supported by the data presented?

-Are the limitations of analysis clearly described?

-Do the authors discuss how these data can be helpful to advance our understanding of the topic under study?

-Is public health relevance addressed?

Reviewer #1: The main limitation of the study is the lack of targeted pressure relief. This is not mentioned as a limitation in the study.

Reviewer #2: -Are the conclusions supported by the data presented?

Yes.

-Are the limitations of analysis clearly described?

Yes.

-Do the authors discuss how these data can be helpful to advance our understanding of the topic under study?

No. They can add this.

-Is public health relevance addressed?

Yes.

Reviewer #3: The conclusion is well formatted, but I suggest listing the main clinical parameters that made it not statistically significant compared to the control group. Example: Healing time? Complete healing? High cost in the study group?

**Editorial and Data Presentation Modifications?**

Reviewer #1: (No Response)

Reviewer #2: (No Response)

Reviewer #3: There is a need for small adjustments, mainly in the methodology between the end of 42 days and 70 days

**Summary and General Comments**

Reviewer #1: The major revision required would be to offload the ulcer with a splint, total contact cast or alternative offloading device. Only then can you conduct the study to reliably determine the effectiveness of the Autologous blood products vs saline moistened gauze.

Reviewer #2: This study is a carefully conducted randomized controlled trial of the use of L-PRF in treating wounds of patients with leprosy in Nepal, with 130 participants enrolled. I congratulate the study team in conducting this study. The study showed that any benefit from L-PRF appeared insufficient to justify routine use in care of neuropathic ulcers in leprosy. Like the authors stated, I also had an impression that the L-PRF fastens the healing process of ulcers in leprosy. It is an important study to be published to alert those using L-PRF in the treatment of leprosy ulcers. 

General comments:

1. There are differences in PRP and L-PRF with different effects being reported. Throughout the manuscript, the distinction between the two is not clear. It will be helpful for the readers if the authors can describe briefly the differences between PRP and L-PRF.

2. The median time of ulcer at enrollment was 22 weeks (over 5 months). May be the patients were too severe to start. What if we intervene at an earlier stage of ulcer development, e.g., between 1-3 months of ulcer development? Could this have changed the outcome? 

3. While the study was for evaluating the L-PRF, it is presenting that regular wound management including timely dressing change even using normal saline and Vaseline is effective in treating leprosy ulcers – this is a comment.

4. I found the use of pedometer in activity monitoring to be an interesting method in removing the bias from ambulatory behavior – this is a comment.

Minor comments:

1. Line 53: Revise ‘n-130’ to ‘n=130’

2. Line 73: The authors mention that they performed a systematic review. However, in order to state as such, they need to provide a more robust descriptions of the methods used. I suggest that they avoid stating that they did systematic review as this can be even another study. What I understand is that they did a ‘literature search’ and observed the gap in evidence for autologous blood products. Change ‘systematic review’ to ‘literature search’. …I understand now after reading the Discussion that the authors did conduct a systematic review (lines 465-466). This information should be moved to the Introduction. Additionally, it will be best if this paper will be available for citation at the time of this paper, or at least as a pre-print.

3. Lines 78-81: The authors provide the title of the trial which one can speculate its objective, but I feel that the objective of the study is not explicitly described. 

4. Line 82, 85, etc. : I find the way this is written , ‘L-PRF/PRP’, to be confusing. The authors can just say ‘L-PRF or PRP’. They are two separate treatment methods.

5. Line 100, lines 106-107: ‘the ulcer was not infected…’, ‘their clinician judged them suitable for treatment, i.e., when the ulcer was clear of debris or infection’: What was the definition of ‘infection’? For example, was this diagnosed based on clinical findings or were any lab works done? This needs to be clarified. If the assessment of infection was done clinically, could its misdiagnosis have affected the outcomes?

6. Line 103: It is better to list the serious medical conditions such as diabetes – they are important information. While it is provided in the protocol published in BMC which is cited, most of the readers will not refer to the paper when reading through this paper. 

7. Lines 201-203: ‘There was thus a ‘closed-loop’ between ulcer image capture in Kathmandu and blinded measurement of ulcer area and complete healing in Birmingham’. � This statement was not very clear to me. 

8. Line 228: As PUSH is a tool for pressure ulcer, was it appropriate to use for this study? Or how was it assessed to be appropriate?

9. Line 231-232: ‘Each ulcer was thus measured using two optical systems yielding three measurements in total.’ From the previous and the following descriptions, I understand that there were three measurements, but this statement is confusing. 

10. Discussions: I didn’t understand why the three measurement methods gave different measurements, especially between ARNAZ and PUSH tool. Comparison of these methods are also interesting and important. Could you elaborate on this?

11. While the blind assessment might have mitigated the bias from non-blinding, it still has its own biases. Photo assessment is not perfect as also mentioned in the text, i.e., ‘there were some cases in which the blinded assessor classed the ulcer as healed in one images, as unhealed in the next image’. Amount of light can affect the presentation, it is not always taken in the same manner, redness can be masked in darker skin tones, etc. The limitations that the study team faced with the photo assessment can be provided.

12. Lines 498-499: This information on the videotape should be included in the Methods section. It is a useful material to understand the whole content of the study.

Reviewer #3: 1-The treatment of chronic ulcers is indeed a huge challenge. There are several treatment methods. I believe that in this important manuscript there could be more parallel discussion of randomized studies that use blood derivatives. And make clear the criteria used to analyze satisfaction.

2- I suggest analyzing injury recurrences and readmissions in both groups in the medium and long term

PLOS authors have the option to publish the peer review history of their article (what does this mean?). If published, this will include your full peer review and any attached files.

Reviewer #1: Yes: Dane Hupp

Reviewer #2: Yes: Rie Yotsu

Reviewer #3: Yes: Francisco Mateus Joao
---

## [Editor Report · Decision Letter 1]

20 Mar 2024

Dear Dr Napit,

We are pleased to inform you that your manuscript 'A randomised Trial of Autologous Blood products, leukocyte and platelet-rich fibrin (L-PRF), to promote ulcer healing in LEprosy: the TABLE trial.' has been provisionally accepted for publication in PLOS Neglected Tropical Diseases.

Best regards,

Linda B Adams

Academic Editor

Mathieu Picardeau

Section Editor

---

## [Editor Report · Acceptance letter]

19 Apr 2024

Dear Dr Napit,

We are delighted to inform you that your manuscript, "A randomised Trial of Autologous Blood products, leukocyte and platelet-rich fibrin (L-PRF), to promote ulcer healing in LEprosy: the TABLE trial.," has been formally accepted for publication in PLOS Neglected Tropical Diseases.

Best regards,

Shaden Kamhawi

co-Editor-in-Chief

Paul Brindley

co-Editor-in-Chief
